# A machine learning algorithm to increase COVID-19 inpatient diagnostic capacity

David Goodman-Meza[1]☉*, Akos Rudas[2,3]☉, Jeffrey N. Chiang[2]☉, Paul C. Adamson[1], Joseph Ebinger[4], Nancy Sun[4], Patrick Botting[4], Jennifer A. Fulcher[1], Faysal G. Saab[5], Rachel Brook[5], Eleazar Eskin[2,6,7], Ulzee An[6], Misagh Kordi[2], Brandon Jew[2], Brunilda Balliu[2], Zeyuan Chen[6], Brian L. Hill[6], Elior Rahmani[6], Eran Halperin[2,6,7,8‡], Vladimir Manuel[9,10‡]

1 Division of Infectious Diseases, David Geffen School of Medicine at UCLA, Los Angeles, California, United States of America, 2 Department of Computational Medicine, UCLA, Los Angeles, California, United States of America, 3 Faculty of Informatics, Eötvös Loránd University (ELTE), Budapest, Hungary, 4 Department of Cardiology, Cedars-Sinai Medical Center, Los Angeles, California, United States of America, 5 Department of Medicine, David Geffen School of Medicine at UCLA, Los Angeles, California, United States of America, 6 Department of Computer Science, UCLA, Los Angeles, California, United States of America, 7 Department of Human Genetics, UCLA, Los Angeles, California, United States of America, 8 Department of Anesthesiology, David Geffen School of Medicine at UCLA, Los Angeles, California, United States of America, 9 Faculty Practice Group, David Geffen School of Medicine at UCLA, Los Angeles, California, United States of America, 10 UCLA Clinical and Translational Science Institute, Los Angeles, California, United States of America

☉ These authors contributed equally to this work.
‡ These authors jointly supervised the work.
* dgoodman@mednet.ucla.edu

**Data Availability Statement:** The datasets generated during and/or analyzed during the current study are not publicly available due to institutional restrictions on data sharing and

## Abstract

Worldwide, testing capacity for SARS-CoV-2 is limited and bottlenecks in the scale up of polymerase chain reaction (PCR-based testing exist. Our aim was to develop and evaluate a machine learning algorithm to diagnose COVID-19 in the inpatient setting. The algorithm was based on basic demographic and laboratory features to serve as a screening tool at hospitals where testing is scarce or unavailable. We used retrospectively collected data from the UCLA Health System in Los Angeles, California. We included all emergency room or inpatient cases receiving SARS-CoV-2 PCR testing who also had a set of ancillary laboratory features (n = 1,455) between 1 March 2020 and 24 May 2020. We tested seven machine learning models and used a combination of those models for the final diagnostic classification. In the test set (n = 392), our combined model had an area under the receiver operator curve of 0.91 (95% confidence interval 0.87–0.96). The model achieved a sensitivity of 0.93 (95% CI 0.85–0.98), specificity of 0.64 (95% CI 0.58–0.69). We found that our machine learning algorithm had excellent diagnostic metrics compared to SARS-CoV-2 PCR. This ensemble machine learning algorithm to diagnose COVID-19 has the potential to be used as a screening tool in hospital settings where PCR testing is scarce or unavailable.

privacy concerns. However, the data can be available from the corresponding author on reasonable request. All code necessary to perform the analyses are available from Zenodo: https://doi.org/10.5281/zenodo.4022238

**Funding:** DGM was supported by the U.S. National Institute on Drug Abuse (K08DA048163, PI Goodman-Meza). PCA was supported by the U.S. National Institute of Mental Health (T32MH080634, PI Currier). JAF was supported by the U.S. National Institute of Allergy and Infectious Disease (K08AI124979, PI Fulcher) and by the Doris Duke Charitable Foundation (Grant 2019086). This project was partially supported by the National Science Foundation (Grant No. 1705197, PI Halperin), and the National Institute of Health (NIH/NHGRI HG010505-02 [PI Halperin], NIH 1R01MH115979 [PI Flint], NIH 5R25GM112625 [PI Eskin], and NIH 5UL1TR001881 [PI Dubinett]). The content is solely the responsibility of the authors and does not necessarily represent the official views of the National Institutes of Health or their employing institutions.

**Competing interests:** No authors have competing interests.

## Introduction

Severe acute respiratory syndrome coronavirus-2 (SARS-CoV2) is a worldwide public health emergency [1, 2]. Polymerase chain reaction (PCR) testing for SARS-CoV-2 is critical to the public health response to coronavirus disease 2019 (COVID-19). PCR testing capacity is especially important in the hospital setting for clinical decision making and infection control procedures [3]. Yet, the inability to scale up testing has been one of the most discussed topics in both the scientific and popular literature [3, 4].

In many hospital settings, PCR testing capacity remains limited. Many PCR assays have short analysis time; however, many hospitals lack on-site PCR capabilities and are tasked with sending samples to centralized laboratories. Transport times and queues lengthen the turn-around time and results can be delayed up to 48 to 96 hours [5–7]. This wait time slows the clinical decision-making process and wastes scarce personal protective equipment.

Machine learning could help fill this gap. Ancillary laboratory values in blood samples of patients with COVID-19 demonstrate a distinct pattern to that of other diseases [3, 8–11]. These changes include elevations in inflammatory markers (ferritin, lactate dehydrogenase [LDH], C-reactive protein, among others) and decreases in certain blood cell counts (absolute lymphocyte count) and an increase in the neutrophil to lymphocyte ratio. Since the SARS-CoV-2 epidemic reached pandemic status, research groups developed prediction algorithms applicable to their particular context [12–16]. One of the major limitations of these previous approaches is that the datasets that were used to train and test the approaches were small. Our aim was to develop a machine learning algorithm using the largest dataset to date, to serve as a COVID-19 diagnostic proxy to be useful in hospitals where SARS-CoV-2 specific PCR testing is unavailable or scarce. We hypothesized that a machine learning-based algorithm based on a parsimonious set of blood markers that include inflammatory markers could predict the presence or absence of COVID-19 with high sensitivity and potentially be used as a screening tool in clinical practice.

## Methods

### Study design

We used electronic health data from the UCLA Health System (Los Angeles, California, USA) to develop a machine learning algorithm to serve as a proxy to diagnose COVID-19 in the hospital setting. Our set of features were selected based on prior studies reporting a difference in these features between patients with and without COVID-19, and higher values in those with severe COVID-19 compared to mild COVID-19 [3, 8–11]. This study was deemed non-human-subjects research by the institutional review board (IRB) at UCLA as all analyses used de-identified data. We report our findings based on STARD-2015 guidelines [17].

### Data sources

We retrospectively considered all cases that were tested for SARS-CoV-2 in the emergency room or inpatient setting within the UCLA Health System between 1 March 2020 and 24 May 2020. After constructing our initial pool of cases, we included only cases with complete blood counts and at least one inflammatory marker (C-reactive protein, ferritin, or LDH) within 48 hours of the sample collection for SARS-CoV2 PCR testing.

All data were extracted from the electronic medical record. Features included in the models were age, gender, hemoglobin, red blood cell count, absolute neutrophil, absolute lymphocyte, absolute eosinophil and absolute basophil counts, the neutrophil to lymphocyte ratio, C-reactive protein, ferritin, and LDH. Prior to entering the model, all features were normalized to have zero mean and unit standard deviation. The normalization parameters (e.g., mean and

standard deviation) were computed using the training set, and the features in the test set were scaled using these values. After scaling, missing lab values were imputed with zero, effectively inserting the mean feature value from the training set. Mean imputation was determined appropriate after evaluating several imputation methods (K-nearest neighbor and Iterative Imputation), which did not result in significant improvements.

### Gold standard

Diagnosis of SARS-CoV-2 was confirmed by PCR testing assays performed at the UCLA Microbiology Laboratory. These assays included the 2019-nCoV Real-Time (RT)-PCR Diagnostic Panel (CDC, Atlanta, GA), the Diasorin Simplexa COVID-19 Direct RT-PCR (Diasorin Molecular LLC, Cypress, CA), the TaqPath COVID-19 Combo Kit (Thermo Fisher Scientific Inc., Waltham, MA).

### Machine learning analysis

We compared seven machine learning models: Random forest, logistic regression, support vector machine, multilayer perceptron (neural network), stochastic gradient descent, XGBoost, and ADABoost. An ensemble (combined) model was then created based on those seven individually trained machine learning models. The final classification as positive or negative was decided using the majority vote of the classifiers calculated by averaging their respective probabilities. The dataset was split 60% for training, 10% for validation, and 30% for testing. The discriminatory operating threshold was determined using a validation set held out from the training set and selected such that the sensitivity on the validation set would be above a predefined threshold of 0.95 by configuring the beta parameter of the F-score. The resulting model was then evaluated on the held-out test set using the following diagnostic metrics: area under the receiver operator curve (AUROC), area under the precision recall curve (AUPRC), sensitivity, specificity, negative predictive value (NPV), and positive predictive value (PPV). Confidence intervals were constructed for each metric using a bootstrapping procedure on the test set in which the test set was repeatedly resampled with replacement 1000 times. Feature importance was assessed using a permutation test on importance. To test the contribution of each feature to model performance, the feature values were randomly shuffled, thereby disrupting their correlations with the outcome, and the decrease in model performance (f1-score) was recorded. All machine learning analyses were performed using Python, making extensive use of the Scikit-learn package.

## Results

### Descriptive

In total, there were 3,444 cases who were tested for SARS-CoV-2 and considered in our analysis. After exclusion of patients who did not have the minimal necessary features to make predictions (a complete blood cell count and at least one inflammatory marker), 1455 cases remained (1273 negative and 182 positive cases) (see Fig 1). All cases were either from the emergency room or inpatient settings. Mean age was 58.1 (SD 22.3), 53% were men, 49% white, 24% Latino, and 29% immunosuppressed. See Table 1 for descriptive characteristics for included features by SARS-CoV-2 status.

### Machine learning model: Diagnostic metrics

The AUROC of the model in the held-out test set (n = 392) was 0.91 (95% confidence interval [CI] 0.87–0.96) and the AUPRC was 0.76 (95% CI 0.66–0.83). The model achieved a sensitivity

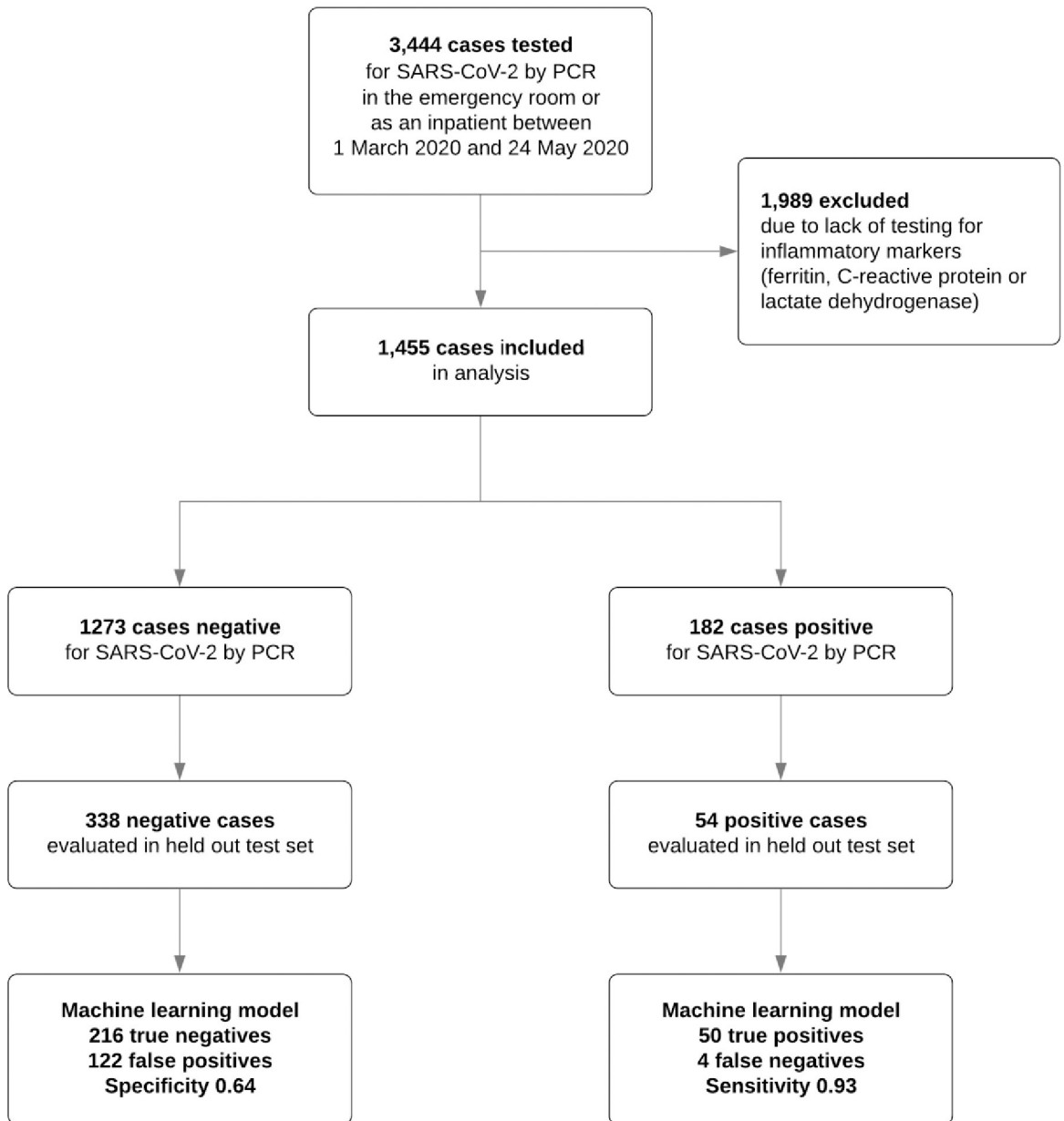

**Fig 1. Diagram of eligible, included and excluded cases, and diagnostic cross tabulation.**

of 0.93 (95% CI 0.84–0.98), specificity of 0.64 (95% CI 0.59–0.69), NPV of 0.98 (95% CI 0.96–1.00), and PPV of 0.29 (95% CI 0.23–0.36). Receiver operator curves and precision-recall curves were presented in Fig 2. Using a feature importance analysis, we found that the features that provide most of the information to the model were: C-reactive protein and LDH (see Fig 3).

In sensitivity analyses, we calculated AUROC and AUPRC when adding the inflammatory features relative to the baseline model of only demographic characteristics and features of the complete blood cell count (see Fig 4). The AUROC of the model of the baseline model was 0.79 (95% CI 0.71–0.85). Then, we added the inflammatory markers to the model one at a time. With ferritin, the AUROC was 0.83 (95% CI 0.78–0.88); with C-reactive protein 0.86 (95% CI 0.79–0.92); with LDH, 0.87 (95% CI 0.82–0.92). The AUPRC of the baseline model

**Table 1. Characteristics of cases by SARS-CoV-2 status.**

| | SARS-CoV-2 status | | | |
| | Negative n (%) | Positive n (%) | Total n | p- value |
|---|---|---|---|---|
| Total | 1273 (87.5) | 182 (12.5) | 1455 | |
| Age, years, mean (SD) | 57.2 (22.6) | 64.2 (19.1) | 58.1 (22.3) | <0.001 |
| Gender | | | | 0.030 |
| Female | 610 (47.9) | 71 (39.0) | 681 (46.8) | |
| Male | 663 (52.1) | 111 (61.0) | 774 (53.2) | |
| Race/ethnicity | | | | 0.006 |
| Asian | 91 (7.1) | 16 (8.8) | 107 (7.4) | |
| Black | 156 (12.3) | 18 (9.9) | 174 (12.0) | |
| Latino | 281 (22.1) | 61 (33.5) | 342 (23.5) | |
| Other | 110 (8.6) | 17 (9.3) | 127 (8.7) | |
| White | 635 (49.9) | 70 (38.5) | 705 (48.5) | |
| Immunosuppressed [+] | 385 (30.2) | 35 (19.2) | 420 (28.9) | 0.003 |
| HIV | 17 (1.3) | 1 (0.5) | 18 (1.2) | 0.590 |
| Transplant | 180 (14.1) | 19 (10.4) | 199 (13.7) | 0.214 |
| Immunosuppressive medications | 312 (24.5) | 29 (15.9) | 341 (23.4) | 0.014 |
| Not immunosuppressed | 888 (69.8) | 147 (80.8) | 1035 (71.1) | |
| Hemoglobin, g/dl, mean (SD) [a] | 11.80 (9.90–13.5) | 12.60 (11.0–14.2) | 11.90 (10.0–13.6) | <0.001 |
| Absolute neutrophil count x 10^3/uL, median (IQR) | 6.02 (3.93–9.39) | 5.19 (3.47–7.46) | 5.92 (3.88–9.12) | 0.001 |
| Absolute lymphocyte count x 10^3/uL, median (IQR) [e] | 1.22 (0.74–1.90) | 0.96 (0.63–1.38) | 1.18 (0.72–1.86) | <0.001 |
| Neutrophil:lymphocyte ratio, median (IQR) | 4.81 (2.47–9.77) | 5.21 (2.91–10.3) | 4.88 (2.56–9.81) | 0.112 |
| Absolute basophil count x 10^3/uL, median (IQR) | 0.03 (0.02–0.05) | 0.01 (0.01–0.03) | 0.03 (0.02–0.05) | <0.001 |
| Absolute eosinophil count x 10^3/uL, median (IQR) | 0.08 (0.02–0.18) | 0.01 (0.00–0.04) | 0.07 (0.01–0.16) | <0.001 |
| Absolute monocyte count x 10^3/uL, median (IQR) | 0.65 (0.47–0.95) | 0.48 (0.33–0.70) | 0.64 (0.44–0.92) | <0.001 |
| Platelet count x 10^3/uL, mean (SD) [b] | 231 (168–298) | 188 (149–252) | 227 (164–291) | <0.001 |
| C-reactive protein, mg/dl, mean (SD) [c] | 1.90 (0.30–7.80) | 6.60 (2.10–12.2) | 2.80 (0.50–8.90) | <0.001 |
| Ferritin, ng/ml, mean (SD) [d] | 216 (93.0–522) | 439 (261–770) | 261 (110–585) | <0.001 |
| Lactate dehydrogenase, U/L, mean (SD) [e] | 245 (192–342) | 306 (231–412) | 261 (198–357) | <0.001 |

**Abbreviations:** IQR, interquartile range; SD, standard deviation.

**Missing values (n, % of total):** [a] hemoglobin 3 (0.2%); [b] platelets 6 (0.4%); [c] C-reactive protein 517 (35.5%); [d] ferritin 737 (50.6%); [e] lactate dehydrogenase 693 (47.6%).

[+] We defined immunosuppressed status as a case with an HIV diagnosis, record of receipt of an organ transplant, or had taken an oral immunosuppressive medication prior to their SARS-CoV-2 test (e.g., prednisone, tacrolimus, mycophenolate, azathioprine, methotrexate).

was 0.50 (95% CI 0.36–0.65); with ferritin 0.56 (95% CI 0.45–0.68); with LDH, 0.66 (95% CI 0.55–0.77); with C-reactive protein 0.66 (95% CI 0.50–0.80). Through these analyses we observed that adding inflammatory markers, especially LDH, CRP, and the combination of the three resulted in statistically significant improvements relative to the baseline model.

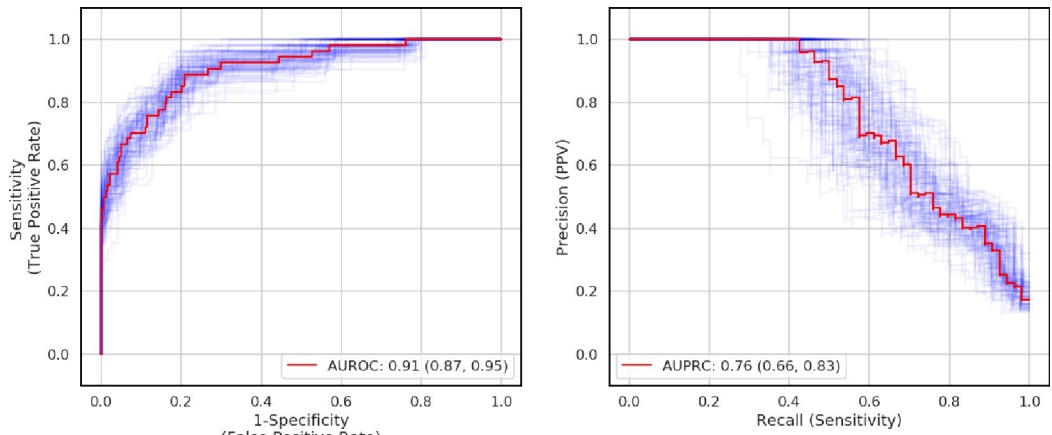

**Fig 2. Performance of the model on the held-out test set (N = 392).** A) Receiver operator curve. B) Precision-recall curve. At a sensitivity-optimized operating threshold, sensitivity and specificity were 0.93 (95% CI 0.85–0.98) and 0.64 (95% CI 0.59–0.69), respectively. Red solid lines were the mean receiver operator curve and mean precision-recall curve, respectively; the purple shaded lines were the curves obtained from the bootstrapping procedure to calculate the 95% confidence intervals.

## Discussion

This is the largest study to date using a machine learning algorithm as a proxy to diagnose COVID-19. We built the algorithm based on a set of basic demographic characteristics and frequently obtained blood biomarkers that could be easily obtained in many hospital settings. Thus, the most likely application of the approach presented in this work is the use of these biomarkers as a proxy for testing in locations where COVID-19 testing is scarce. We showed a high sensitivity for COVID-19 diagnosis when compared to SARS-CoV-2 RT PCR testing as the gold standard. The blood biomarkers included in the model can be obtained with a single blood draw and turnaround time is typically within 24 hours at most hospital centers with laboratory capabilities. Due to the model's high sensitivity and rapid turnaround time, the proposed algorithm lends itself to practical use in hospital facilities as a screening tool. At the time of submission, this model was being actively developed into a web or mobile application, whereby a clinician inputs the obtained values and receives immediate prediction on the probability of a particular patient having COVID-19. Further validation will be required to ascertain its performance in other medical centers.

Our set of features performed as well as, or better than, the three diagnostic algorithms with the largest number of cases known to us at this time [12, 13, 16]. A report by Sun et al. used epidemiologic, clinical, laboratory and imaging features in their algorithms and reported AUROCs of 0.91 (full model), 0.88 (without epidemiologic features), 0.88 (without imaging features), and 0.65 (with clinical features alone) [12]. They used features from a complete blood cell count and from a basic chemistry panel (sodium and creatinine), whereas, we used inflammatory markers (ferritin, C-reactive protein, LDH) instead of sodium, potassium, and creatinine as we did not suspect significant differences *a priori* in sodium, potassium, or creatinine. Meng et al reported an AUROC of 0.89 using a different set of features that included activated partial thromboplastin time, triglycerides, uric acid, albumin/globulin, sodium, and calcium [16]. Batista et al. developed an algorithm aimed for use in lower resource settings and reported an AUROC of 0.87 in a sparser dataset that only included basic demographics and complete blood cell counts [13]. In fact, our model which incorporated inflammatory markers significantly improved upon this set of features in terms of both AUROC and AUPRC. For a

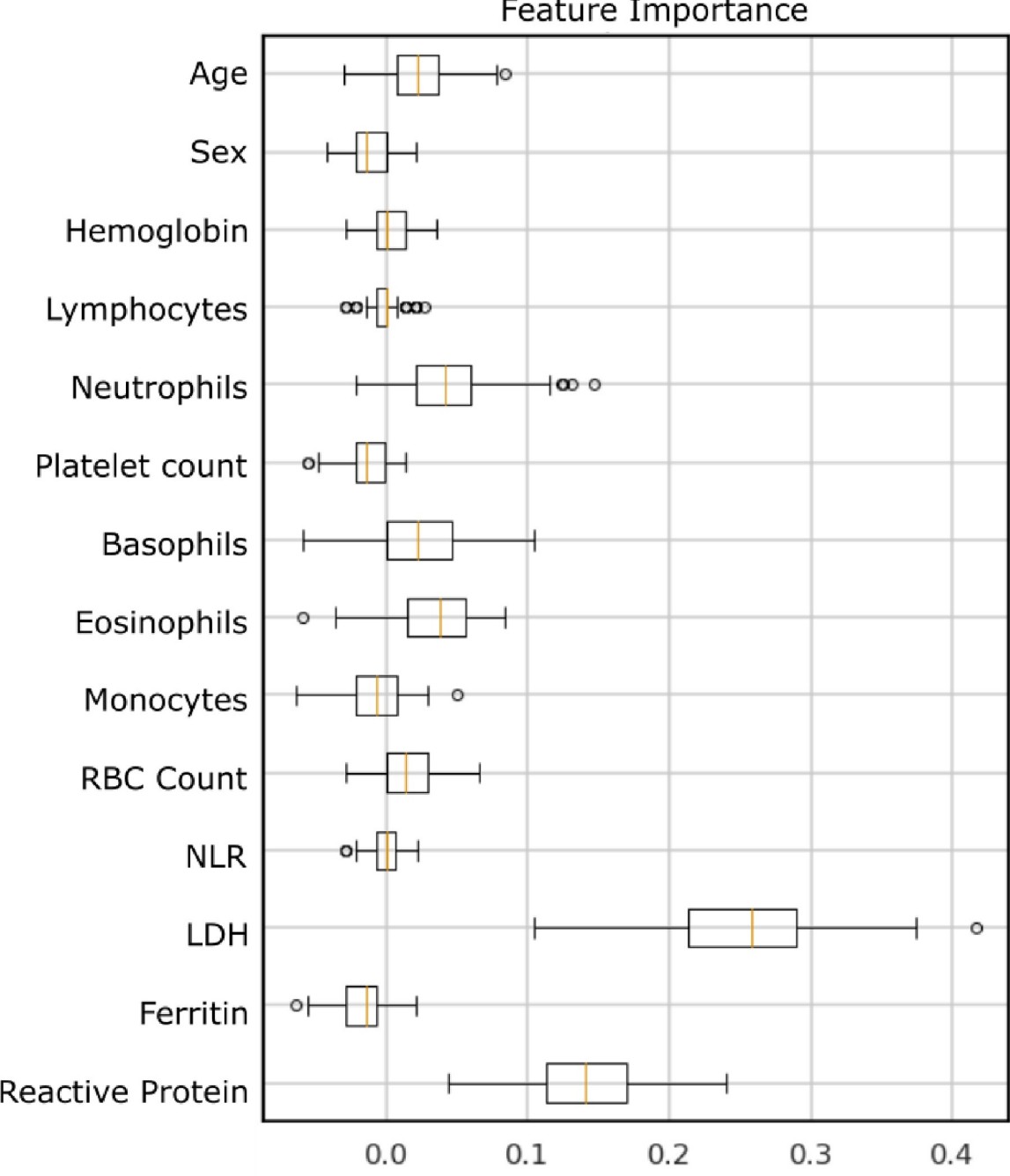

**Fig 3. Combined model feature importance.** Decrease in model performance (f1-score) after randomly shuffling the respective feature values. Higher values represent important features for classification. Abbreviations: LDH, lactate dehydrogenase; NLR, neutrophil to lymphocyte ratio; RBC, red blood cells.

full comparison of diagnostic algorithms related to COVID-19 we refer the reader to [15]—a living systematic review.

Our findings should be considered in light of the following limitations. We included data from one medical center in Los Angeles. Incorporating data from other medical centers in other geographic areas would provide a higher likelihood of generalizability. Second, although

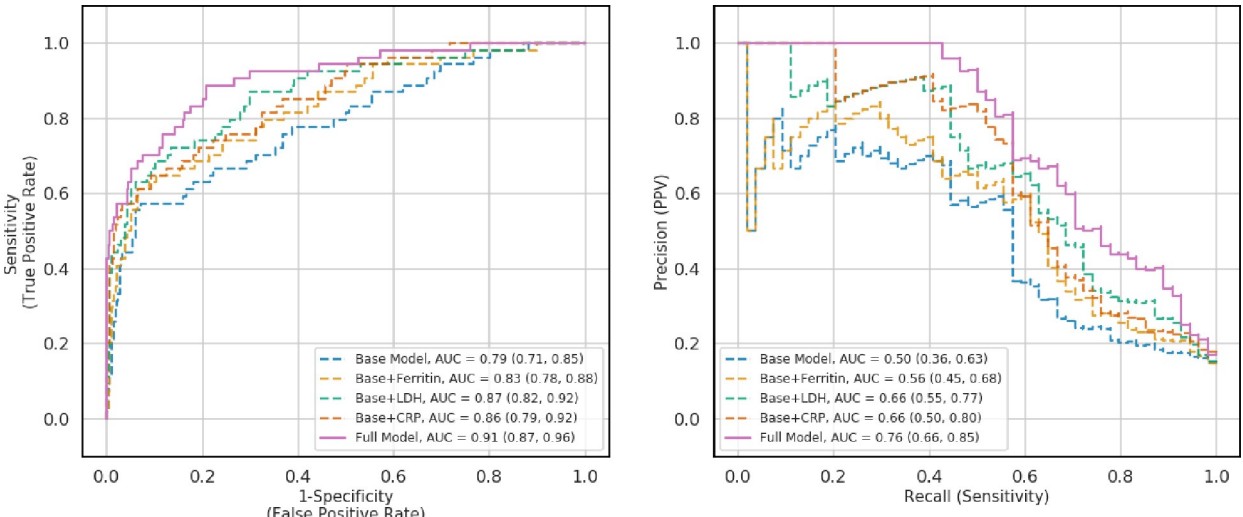

**Fig 4. Performance of models while removing one of the features.** All analyses were performed on the held-out test set (N = 392). A) Receiver operating curve. B) Precision-recall curve. Base model includes only demographic features and complete blood cell count. Abbreviations: CRP, C-reactive protein; LDH, lactate dehydrogenase.

many of our patients either had immunosuppressive conditions (e.g., solid organ transplants) or were taking immunosuppressive medications (e.g., steroids), immunosuppressed hosts are a heterogenous group and their immunosuppression may impact the laboratory values we used in our models. We would need more cases with those conditions to understand how the algorithm would perform in these populations. It is likely that specific models tailored to the immunocompromised host should be developed to improve accuracy in this population. Third, it is also possible that other community respiratory viral infections (e.g., influenza, RSV) could cause a similar laboratory profile; however, incidence of these other community respiratory viruses was low during the case inclusion period. Further validation comparing COVID-19 cases to cases of other community respiratory viruses is needed. Finally, as all of our patients' blood was tested in the emergency department or as an inpatient, the applicability of this model in the outpatient setting or milder cases of COVID-19 is unclear.Our report, in combination with others [12, 13, 15, 16], demonstrate the high diagnostic accuracy of machine learning models based on early available data. Other models have also been developed based on characteristic imaging changes [15]. We and others were able to demonstrate impressive results in our data silos [12–15]. Yet, to realize the full potential of machine learning and its applicability to clinical medicine, collaborations from the international community are crucial, both for the sharing of data and for the development and validation of advanced algorithms. It is unclear if testing capacity for active disease using PCR-based methods will ever meet the expanding need globally. In fact, countries in low-resource settings, such as in Sub-Saharan Africa or Latin America, face bottlenecks in the testing supply chain, and are unable to compete with affluent nations for prohibitively expensive PCR test kits. Even in developed nations, scale up of PCR-based testing has many bottlenecks that include purchase of new testing platforms, sample acquisition, availability of reagents, swabs and transport media, and the technical human expertise in performing PCR tests.

In summary, by using readily available laboratory tests combined with machine learning we achieved a high sensitivity comparable to that of PCR. This machine learning modality may be especially useful as a screening test in smaller medical centers or those in resource-poor regions that may have limited capacity for COVID-19 PCR-based diagnosis, or in instances

were testing capacity is in danger due to low supplies. Further validation is necessary in diverse geographic settings and in a prospective manor to be used is a reliable tool to support clinical decision making.

## Supporting information

**S1 Checklist.**
(PDF)

## Author Contributions

**Conceptualization:** David Goodman-Meza, Paul C. Adamson, Jennifer A. Fulcher, Faysal G. Saab, Rachel Brook, Vladimir Manuel.

**Data curation:** Joseph Ebinger, Nancy Sun, Patrick Botting.

**Formal analysis:** Akos Rudas, Jeffrey N. Chiang, Ulzee An, Misagh Kordi, Brandon Jew, Brunilda Balliu, Zeyuan Chen, Brian L. Hill, Elior Rahmani, Eran Halperin.

**Methodology:** David Goodman-Meza, Eran Halperin, Vladimir Manuel.

**Software:** Misagh Kordi.

**Supervision:** Eleazar Eskin, Eran Halperin, Vladimir Manuel.

**Writing – original draft:** David Goodman-Meza.

**Writing – review & editing:** David Goodman-Meza, Paul C. Adamson, Joseph Ebinger, Jennifer A. Fulcher, Faysal G. Saab, Rachel Brook, Eleazar Eskin, Eran Halperin, Vladimir Manuel.

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
