## [Decision Letter · Decision Letter 0]

8 Sep 2020

A m achine learning algorithm to increase COVID-19 inpatient diagnostic capacity

PONE-D-20-20126

Dear Dr. Goodman-Meza,

We’re pleased to inform you that your manuscript has been judged scientifically suitable for publication and will be formally accepted for publication once it meets all outstanding technical requirements.

Kind regards,

Ryan J. Urbanowicz, Ph.D.

Academic Editor

PLOS ONE

Additional Editor Comments (optional):

Reviewers' comments:

Reviewer's Responses to Questions

**Comments to the Author**

1. Is the manuscript technically sound, and do the data support the conclusions?

Reviewer #1: Partly

Reviewer #2: Yes

2. Has the statistical analysis been performed appropriately and rigorously? 

Reviewer #1: Yes

Reviewer #2: Yes

3. Have the authors made all data underlying the findings in their manuscript fully available?

Reviewer #1: No

Reviewer #2: No

4. Is the manuscript presented in an intelligible fashion and written in standard English?

Reviewer #1: Yes

Reviewer #2: Yes

5. Review Comments to the Author

Reviewer #1: * Yes, the manuscript is technically sound. The only reason I said "Partly" to question 1 above is that for reasons out of the authors' control, the sample size is somewhat small. However, as they say, there is the (very exciting) prospect that such work can lead to data-sharing, particularly among hospitals in diverse regions for data-diversity, leading to much larger training data-sets and hence, more-accurate models for realistic data.

* The statistical analysis is rigorous.

* The answer to question 3 is "No" since the data is anonymized medical-center data: the authors say "The datasets generated during and/or analyzed during the current study are not publicly available due to institutional restrictions on data sharing and privacy concerns. However, the data can be available from the corresponding author on reasonable request. All code necessary to perform the analyses will be available on a public repository by the time of publication." I think this is reasonable, and really the best that can be expected under the circumstances.

* The paper is well-written in general, but I have one technical question (see (e) below) and a few minor comments:

(a) Page 9: "pandemic status research" -> pandemic status, research"

(b) Hyphenate multi-word adjectives throughout: e.g., as in "machine-learning algorithm"

(c) Page 10: "from UCLA Health System" -> "from the UCLA Health System"

(d) Page 10: "non-human subjects" -> "non-human-subjects"

(e) Page 10, on normalizing all features: How do you do such normalization for gender -- a discrete feature with very small support?

Reviewer #2: The paper by Goodman-Meza et al. describes an ensemble machine learning algorithm for the diagnosis of COVID-19. Specifically, using the largest available dataset of patients testing for COVID-19 in the hospital setting, the authors make use of demographic and laboratory features to obtain highly accurate predictions. Their results are comparable to the gold-standard PCR test. This work is particularly valuable for COVID-19 diagnosis at hospitals with limited resources or where standard testing is too slow. Despite the limitation of generalizability, overall this research provides a useful model for the diagnosis of COVID-19 in the hospital setting.

Patients were excluded from the analysis if they did not have a CBC and at least one inflammatory lab value. Patients who tested positive for COVID-19 were more likely to be older, male, and not immunosuppressed. Drawing upon recent literature, the authors chose age, gender, seven features from blood cell counts, and three inflammatory markers as features for their model. Missing data was imputed post-normalization using the mean values.

Using seven machine learning models, the authors created an ensemble machine learning algorithm which classified patients as positive or negative using a majority vote. The data was split 60/10/30 for training, validation, and testing, as is standard for many machine learning analyses, the. AUROC, AUPRC, NPV, PPV, sensitivity and specificity were reported on the testing set, in addition to confidence intervals generated by bootstrapping. Two of the inflammatory markers, LDH and C-reactive protein, exhibited the highest feature importance using a permutation test. Sensitivity analyses further demonstrated the utility of these inflammatory markers as additions to the baseline model. The authors make note of the limitations surrounding the generalizability to outpatient settings and the fairly high number of immunocompromised patients among the cases.

Key strengths:

1. This paper drew on previous literature to choose the most informative laboratory features to include in the model. All of the features in the model are commonly and easily obtained in the hospital setting, even in resource-poor areas.

2. The authors reported a variety of related performance metrics that all demonstrated the algorithm’s ability to capture true positives and limit false negatives. This model performed very well, notable demonstrated by the AUROC, sensitivity, and NPV. Prioritizing sensitivity and NPV are of keen clinical importance in this context and especially during a pandemic.

3. Finally, the authors used Python’s Scikit-learn package to perform the machine learning analyses, a highly accessible and open source software. They have agreed to make all of their code public and are in the process of creating a wed/mobile application for expanded use.

Suggestions for improvement:

1. Nearly 2000 patients who were tested for COVID-19 were excluded from this analysis due to incomplete laboratory measures. It would be helpful to provide some information or comments on why these patients did not have these particular lab measures, and if this may result in selection bias (for more severe cases). For example, were these excluded patients more likely to be negative or have mild cases of COVID-19? At a minimum, this limitation should be acknowledged.

2. Relatedly, under study design in the Methods section, the authors note that their features were selected in part based on higher values in those with severe COVID-19. This should also be noted in the limitations, given that generalizability to patients with milder cases of COVID-19 is unclear.

3. The authors provided no explanation for their choice of machine learning (ML) algorithms. Granted, the seven listed methods are all classification methods within Scikit-learn’s supervised machine learning models. It would nevertheless be useful to include a rationale or references for why these seven were specifically chosen.

4. Lastly, the authors stated that “We compared seven machine learning models…” (Methods, Machine learning analysis) but provided no data, figures, or discussion on this ‘comparison’. An explanation or figure summarizing individual model performance would provide additional clarity to this statement.

6. PLOS authors have the option to publish the peer review history of their article (what does this mean?). If published, this will include your full peer review and any attached files.

Reviewer #1: No

Reviewer #2: No

---

## [Editor Report · Acceptance letter]

14 Sep 2020

PONE-D-20-20126 

A machine learning algorithm to increase COVID-19 inpatient diagnostic capacity 

Dear Dr. Goodman-Meza:

I'm pleased to inform you that your manuscript has been deemed suitable for publication in PLOS ONE. Congratulations! Your manuscript is now with our production department. 

Kind regards, 

on behalf of

Dr. Ryan J. Urbanowicz 

Academic Editor

PLOS ONE